# Autofocusing of Maneuvering Targets in Terahertz Inverse Synthetic Aperture Radar Imaging Based on Damped Newton Method

**DOI:** 10.3390/s22186883

**Published:** 2022-09-12

**Authors:** Hetong Wang, Qi Yang, Hongqiang Wang, Bin Deng

**Affiliations:** College of Electronic Science, National University of Defense Technology, Changsha 410073, China

**Keywords:** terahertz radar, minimum entropy, inverse synthetic aperture radar (ISAR), autofocusing, damped newton method

## Abstract

Maneuvering target imaging based on inverse synthetic aperture radar (ISAR) imaging has recently drawn significant attention. Among the many autofocusing technologies which are crucial in ISAR imaging, minimum-entropy-based autofocusing (MEA) is highly robust. However, traditional MEA is not suitable for terahertz (THz) ISAR imaging. For one thing, the iterative process in traditional MEA is too complicated to be utilized for THz-ISAR imaging with tremendous data. For another, THz wavelengths are very short and extremely sensitive to phase errors, so the compensation accuracy of the traditional MEA method can hardly meet the requirements of THz radar high-resolution imaging. Therefore, in this paper, the MEA algorithm based on the damped Newton method is proposed, which improves computational efficiency by approximating the first- and second-order partial derivatives of the image entropy function with respect to the phase errors, as well as by the fast Fourier transform (FFT). The search step size factor is introduced to ensure that the algorithm can converge along the declination direction of the entropy function and obtain the globally optimal ISAR image. The experimental results validated the efficiency of the proposed algorithm, which is promising in THz-ISAR imaging of maneuvering targets.

## 1. Introduction

Inverse synthetic aperture radar (ISAR) enables the image of noncooperative targets with high resolution without the restrictions of illumination and weather and has been widely applied in many military and civil fields, such as space surveillance, air control, missile defense, air defense, and so on [1,2,3,4]. Until now, the research on maneuvering target imaging has mainly focused on low frequency bands, such as below the W-band. However, owing to the low resolution of low-frequency radar imaging, it is difficult to recognize and interpret the imaging results in this band. Compared with low-frequency radar, Terahertz (THz) radars can more easily obtain a higher carrier frequency and wider absolute bandwidth [5,6,7], which can realize high-resolution imaging in a very short observation time and capture high-speed moving target dynamics in real-time, avoiding the complex imaging process caused by the rapid changes of target attitude [8,9,10,11,12,13,14]. Furthermore, THz-ISAR imaging has great potential in the detection and identification of maneuvering targets, which is of great significance to be further researched and developed [15].

Because ISAR imaging is mainly applied to image noncooperative moving targets (aircraft, missiles, warships, satellites, etc.), the prior information of the target cannot be obtained before imaging, so the image quality is mainly determined by the performance of the motion compensation of the radar echoes. The translational motion compensation is necessary for generating high-resolution THz-ISAR images. It is usually accomplished through two steps: range alignment and autofocusing. The accuracy requirement of range alignment is relative to the range resolution [16,17], and the accuracy requirement of autofocusing is relative to the radar wavelength [18]. Compared with range alignment, the requirement of autofocusing accuracy is more stringent, as the focus and sharpness of the ISAR image mainly depend on the autofocusing accuracy. Such a problem is more difficult in the THz band, as the shorter wavelength puts forward higher requirements for autofocusing accuracy [15]. To the best of our knowledge, limited research about ISAR imaging of maneuvering targets with THz radars is available in the current literature. 

Autofocusing methods can be generally divided into parametric and nonparametric methods [19]. The parametric method aims to reconstruct the motion trajectory of the target and requires accurate modeling of the phase error. In practical applications, the radar echo is seriously affected by noise [1]. Hence, the established model usually has difficulty accurately simulating the phase error caused by target translational motion [19]. Nonparametric methods make no hypothesis of the phase error and are more applicable than parametric methods. Therefore, many nonparametric autofocusing algorithms have been proposed, including the dominant scatterer algorithm (DSA), multiple-scatterer algorithm (MSA) [20], Doppler centroid tracking (DCT) algorithm [21], and phase gradient algorithm (PGA) [22]. A series of algorithms based on the target’s dominant scatterers, such as DSA and MSA, is suitable for situations where there are isolating dominant scatters on the target. However, it is difficult to determine the range bin where isolating dominant scatterers are located in some practical cases, which seriously affects the imaging quality [19]. DCT tracks the entire target rather than a single scattering point, so its performance is relatively stable. However, this method suffers from low compensation accuracy [19]. The PGA eliminates the influence of translational phase components through the steps of cyclic shift, isolation, and iteration in the frequency domain, but its accuracy is limited by the window width [23]. There are some maximum contrast-based methods [24,25] that present faster convergence, but these methods depend on dominant scatterers so much that most scatterers may not be as well focused as the dominant one. Liu [26] proposed an eigenvector-based autofocus method using the minimum-entropy-based method for range alignment. This subspace-based method utilizes range-aligned data to construct the covariance matrix to estimate phase error and has good performance at low signal-to-noise ratios (SNRs), but with high computational complexity. 

Another type of nonparametric autofocusing algorithm takes the minimum entropy as global quality indices for an ISAR image and uses the phase error of each pulse as an unknown variable to perform an iterative search to improve the estimation precision of the phase errors caused by the translational motion, thus having better performances in imaging maneuvering targets [25]. However, due to the restriction of imaging efficiency by the high sampling rate and the large amount of data of the THz radar imaging system, the number of iterative searches during calculation and data processing are significantly important factors that need to be considered. Li [27] proposed a minimum-entropy-based autofocusing (MEA) method based on the stage-by-stage approaching (SSA) algorithm. This algorithm is implemented with a downhill searching technique, the intolerable computation efficiency of which is not suitable for THz-ISAR imaging with a large amount of data. Wang [28] proposed a fixed-point MEA (FP-MEA) algorithm, which derives an iterative solution to obtain the phase error by minimizing the image entropy. Although the FP-MEA method is more computationally efficient than SSA-MEA, it is difficult for the phase compensation accuracy to meet the demand of THz radar high-resolution imaging. Zhang [19] proposed the Newton-method-based minimum entropy algorithm (N-MEA) to improve computational efficiency. However, if the initial value of the N-MEA is too far from the minimum point, it leads to a situation where the function slowly converges or even fails to converge. To ensure the convergence of the image entropy, initialization iteration is performed using DCT. Moreover, the method uses fixed-step iterations, which sometimes make the value of the entropy function rise and do not guarantee a stable decrease of image entropy in dealing with the optimization problems with non-quadratic objective functions [29,30]. Wang [23] proposed a method based on MEA and an adaptive modified Fourier transform (MFT). The approach combines phase autofocus and azimuth compression; it is difficult for the computational efficiency of this method to meet the needs of the large datum volume of terahertz radar. Table 1 shows the comparison of several popular motion compensation methods at present. These methods have limited application scenarios.

Taking the robustness, adaptability, and precision into consideration, damped-Newton-method-based MEA (DN-MEA) was proposed to meet the requirements of a large amount of data and high-resolution imaging of THz radar. The main contributions of this paper are as follows.

(1)The proposed method did not need to model the trajectory of the target and had strong adaptability. The phase error of the target was estimated by minimizing the entropy of the ISAR image in each iteration.(2)Since the phase errors between pulses are independent of each other, the simplified Hessian matrix was used to improve the calculation efficiency. The variable step-size factor, introduced to avoid the entropy function falling into the local optimal solution, was decided by the golden section method [29], which ensured the stable decline of image entropy and further improved the robustness of MEA under a low SNR.(3)A large number of experimental results based on the measured data of 0.32 THz radar verified the effectiveness and robustness of the proposed DN-MEA method.

This paper is organized as follows: Section 2 presents the signal model after envelope alignment. Section 3 proposes the ISAR autofocusing algorithm based on the damped Newton method. Section 4 analyzes the experimental results of the measured data. A conclusion is given in Section 5.

## 2. Signal Model for ISAR Imaging

The equivalent diagram of ISAR translational motion is shown in Figure 1. The motion of the maneuvering target can be divided into three parts: the first part is a circular motion with radius $R_{0}$. The second part is the translational motion along the radar line of sight (LOS). The third part is the rotational motion relative to the reference point. The first part did not affect the imaging process, and the second part, as the translational motion component, had such a great impact on the quality of the ISAR image that it could cause one scatterer to spread in multiple range cells. In this way, the echo phases of the scattered electric field data at the scatter point could not be aligned between pulses, so the Doppler frequencies used to estimate the azimuthal position of the scatter points were spread across multiple range bins. This caused the scatterer image to be defocused in the final image. Therefore, it was of vital importance to compensate for the translational motion component. The third part is the change of the azimuth angle of the radar to the target, which made it possible for ISAR to distinguish the scattering points.

We estimated the phase error caused by the target translational motion, and the effect of the target’s translational motion on the phase and ISAR image was investigated based on the geometry as illustrated in Figure 2.

P(x,y) represents the scattering points on the target. The distance of point P from the radar can be expressed as
(1)r(t)=R(t)+xcosωt−ysinωt
where R(t) represents the translational motion that needed to be compensated and ω denotes the rotational speed of the target with respect to the radar LOS axis u. Expanding R(t) into their Taylor series, it yields
(2)$$R(t)=R_{0}+v_{t}t+\frac{1}{2}a_{t}t^{2}+\cdots$$
where $R_{0}$ represents the initial range of the target, and $v_{t}$ and $a_{t}$ are the target’s translational velocity and acceleration, respectively. The Doppler frequency shift can be expressed as
(3)$$\begin{array}{cc} f_{D} & =-\frac{2f_{c}}{c}\frac{d}{\mathrm{dt}}r(t)\\ & =-\frac{2f_{c}}{c}\left[v_{r}+a_{t}t-\omega x\sin\omega t-\omega y\cos\omega t+\cdots \right]\\ & \approx -\frac{2f_{c}}{c}(v_{r}+a_{t}t-x\omega^{2}t-\omega y+\cdots )\\ & =f_{D,trans}+f_{D,rot} \end{array}$$

We applied the approximation in Equation (3):(4)$$\begin{array}{l} \cos\omega t\approx 1\\ \sin\omega t\approx \omega t \end{array}$$
where $f_{D,trans}=-\frac{2f_{c}}{c}(v_{r}+a_{t}t+\cdots )$ represents the Doppler shift due to translational motion, which caused the image to be defocused or even unable to be imaged;$f_{D,rot}=\frac{2f_{c}}{c}(x\omega^{2}t+\omega y)$ represents the Doppler shift due to rotational motion, where the first term caused the image to defocus and the second term contributed to imaging. The influence of translational motion on imaging quality was much greater than that of rotational motion. $f_{c}$ denotes the carrier frequency, where the translational Doppler shift became directly related to the carrier frequency $f_{c}$. Due to the high carrier frequency of THz radar, the Doppler frequency greatly changed during the imaging time. If the phase error could not be well compensated, the imaging would not be completed. The process of compensating for phase errors is called autofocusing. The premise of autofocusing is that the envelopes of the echoes have been aligned. Many methods have been proposed to achieve envelope alignment, including the minimum entropy [17] and envelope correlation methods [34]. The formula derivation before envelope alignment was omitted [1]. Assuming that the envelopes of the echoes are aligned, the aligned one-dimensional range profiles can be expressed as f(m,n). Autofocusing and ISAR imaging can be formulated as [28]:(5)$$I(k,n)=\sum_{m=0}^{M-1}f(m,n)\exp(j\varphi_{m})\exp(-j\frac{2\pi }{M}km)$$
where k, m, and n are the indices of Doppler frequency, slow time, and range bins, respectively. I(k,n) represents the phase-adjusted ISAR image and $\varphi_{m}$ represents the phase error to be adjusted. The phase error was only related to slow time m and had nothing to do with the range bins n. That is to say, for all range cells, the initial phase error of the same azimuth bin kept the same. The entropy of the two-dimensional image, which was used to measure the focus quality of the ISAR image, is written as
(6)$$E_{I}=-\sum_{k=0}^{M-1}\sum_{n=0}^{N-1}\frac{{\left|I(k,n)\right|}^{2}}{S}\frac{S}{\ln{\left|I(k,n)\right|}^{2}}$$

The total energy S of the radar image is a constant, the value of which is
(7)$$S=\sum_{k=0}^{M-1}\sum_{n=0}^{N-1}{\left|I(k,n)\right|}^{2}$$

S is a phase-independent constant, so the image entropy can be rewritten as
(8)$$E_{I}{}^{\prime }=-\sum_{k=0}^{M-1}\sum_{n=0}^{N-1}{\left|I(k,n)\right|}^{2}\ln{\left|I(k,n)\right|}^{2}$$

The optimal phase compensation factor of the signal h(m,n) can be expressed as follows: (9)$${\overset{\wedge }{\varphi }}_{m}=\arg\{\underset{\varphi_{m}}{\min}[E_{I}{}^{\prime }]\},m=0,1,\ldots ,M-1$$
which means the $\varphi_{m}$ that minimizes $E_{I}{}^{\prime }$ satisfies
(10)$${\frac{\partial E_{I}{}^{\prime }}{\partial \varphi_{m}}|}_{\varphi_{m}=\overset{\wedge }{\varphi_{m}}}=0$$

The correction phase can be estimated through an iterative process. The condition for the end of the iteration was $\underset{m}{\max}\left\{\left|\varphi_{m}^{l+1}-\varphi_{m}^{l}\right|\right\}\leq \mu$, and μ is a constant, indicating the accuracy of the phase compensation.

## 3. The Proposed Methodology

In the ISAR imaging process, the image entropy can be regarded as a function of $\varphi_{m}$, and the minimum value of entropy can be searched along the negative direction of the gradient to obtain the estimated value of $\varphi_{m}$. The Newton iterative method with a second-order convergence rate is usually used to solve optimization problems. When using the Newton method to solve Equation (9), the Hessian matrix is required to be positive definite, and the initial value of iteration should be selected close enough to the optimal value to ensure local convergence. When estimating the phase error, this requirement cannot be guaranteed. Therefore, in this paper, the damped Newton method was innovatively applied to solve Equation (9) to ensure that the image entropy could consecutively decrease along the correct iterative direction. Compared with [19], the proposed method allowed the entropy function to converge to a minimum without setting the initial value of the iteration in advance.

The derivative of $E_{I}{}^{\prime }$ with respect to $\varphi_{m}$ can be represented as follows [28]:(11)$$\frac{\partial E_{I}{}^{\prime }}{\partial \varphi_{m}}=-\sum_{k=0}^{M-1}\sum_{n=0}^{N-1}\left(1+\ln{\left|I(k,n)\right|}^{2}\right)\frac{\partial {\left|I(k,n)\right|}^{2}}{\partial \varphi_{m}}$$

Because ${\left|I(k,n)\right|}^{2}=I(k,n)I^{*}(k,n)$, therefore: (12)∂|I(k,n)|2∂φm=I(k,n)∂I*(k,n)∂φm+I*(k,n)∂I(k,n)∂φm=2Re{I*(k,n)∂I(k,n)∂φm}
where $\frac{\partial I(k,n)}{\partial \varphi_{m}}$ can be obtained from Equation (5):(13)$$\frac{\partial I(k,n)}{\partial \varphi_{m}}=j\cdot f(m,n)\exp(j\varphi_{m})\exp(-j\frac{2\pi }{M}km)$$

Combining Equations (11) and (12) with Equation (13), the $\frac{\partial E_{I}{}^{\prime }}{\partial \varphi_{m}}$ can be rewritten as [28]:(14)∂EI′∂φm=2Im{∑n=0N−1H1*(m,n)f(m,n)exp(jφm)}

$H_{1}(m,n)$ can improve computational efficiency by an inverse fast Fourier transform (IFFT), which is represented as follows:(15)$$\begin{array}{cc} H_{1}(m,n) & =\sum_{k=0}^{M-1}\left[1+\ln{\left|I(k,n)\right|}^{2}\right]\cdot I(k,n)\exp(j\frac{2\pi }{M})km\\ & =IFFT\left\{\left[1+\ln{\left|I(k,n)\right|}^{2}\right]I(k,n)\right\} \end{array}$$

The Hessian of $E_{I}{}^{\prime }$ with respect to $\varphi_{m}$ is defined as
(16)$$\partial^{2}E_{I}{}^{\prime }=\left[\begin{array}{cccc} \frac{\partial^{2}E_{I}{}^{\prime }}{\partial \varphi_{1}^{2}} & \frac{\partial^{2}E_{I}{}^{\prime }}{\partial \varphi_{1}\partial \varphi_{2}} & \cdots & \frac{\partial^{2}E_{I}{}^{\prime }}{\partial \varphi_{1}\partial \varphi_{m}}\\ \frac{\partial^{2}E_{I}{}^{\prime }}{\partial \varphi_{2}\partial \varphi_{1}} & \frac{\partial^{2}E_{I}{}^{\prime }}{\partial \varphi_{2}^{2}} & \cdots & \frac{\partial^{2}E_{I}{}^{\prime }}{\partial \varphi_{2}\partial \varphi_{m}}\\ \vdots & \vdots & \ddots & \vdots \\ \frac{\partial^{2}E_{I}{}^{\prime }}{\partial \varphi_{m}\partial \varphi_{1}} & \frac{\partial^{2}E_{I}{}^{\prime }}{\partial \varphi_{m}\partial \varphi_{2}} & \cdots & \frac{\partial^{2}E_{I}{}^{\prime }}{\partial \varphi_{m}^{2}} \end{array}\right]$$

Since the phase errors of the different pulses of ISAR are relatively independent, only the diagonal elements of the Hessian matrix were derived to simplify the calculation:(17)$$\partial^{2}E_{I}{}^{\prime }=\left[\begin{array}{cccc} \frac{\partial^{2}E_{I}{}^{\prime }}{\partial \varphi_{1}^{2}} & 0 & \cdots & 0\\ 0 & \frac{\partial^{2}E_{I}{}^{\prime }}{\partial \varphi_{2}^{2}} & \cdots & 0\\ \vdots & \vdots & \ddots & \vdots \\ 0 & 0 & \cdots & \frac{\partial^{2}E_{I}{}^{\prime }}{\partial \varphi_{m}^{2}} \end{array}\right]$$

The diagonal elements of the Hessian matrix $\frac{\partial^{2}E_{I}{}^{\prime }}{\partial \varphi_{m}^{2}}$ can be expressed as follows. The detailed derivation process of $\frac{\partial^{2}E_{I}{}^{\prime }}{\partial \varphi_{m}^{2}}$ is given in Appendix A:(18)∂2EI′∂φm2=−2∑k=0M−1∑n=0N−1(2+ln|I(k,n)|2)⋅|f(m,n)|2+2Re{∑n=0N−1[∑n=0M−1[1+ln|I(k,n)|2]⋅I(k,n)exp(j2πM)km]f(m,n)exp(jφm)}

Then, the iterative process of DN-MEA can be expressed as
(19)$$\varphi_{m}^{l+1}=\varphi_{m}^{l}-\beta_{l}{\left[{\left(\frac{\partial^{2}E_{I}{}^{\prime }}{\partial \varphi_{m}^{2}}\right)}^{-1}\frac{\partial E_{I}{}^{\prime }}{\partial \varphi_{m}}\right]|}_{\varphi_{m}=\varphi_{m}^{l}}$$
where $p_{l}=-{\left[{\left(\frac{\partial^{2}E_{I}{}^{\prime }}{\partial \varphi_{m}^{2}}\right)}^{-1}\frac{\partial E_{I}{}^{\prime }}{\partial \varphi_{m}}\right]|}_{\varphi_{m}=\varphi_{m}^{l}}$ is the search direction and the factor $\beta_{l}$ represents the search step, which could make $\varphi_{m}^{l}$ move in the search direction $p_{l}$ until a sufficient decrease of entropy was reached. $\beta_{l}$ satisfied the following conditions:(20)$$E_{I}{}^{\prime }(\varphi_{m}^{l}+\beta_{l}p_{l})\leq \underset{\beta \geq 0}{\min}E_{I}{}^{\prime }(\varphi_{m}^{l}+\beta p_{l})$$

$\beta_{l}$ can be determined by the line search method [30]. In order to prevent the entropy function from falling into the local optimal solution, the advance-and-retreat method was used to estimate the interval of $\beta_{l}$, and then the golden section method was used to evaluate the value of $\beta_{l}$. A flow chart is given in Figure 3 to elucidate DN-MEA.

## 4. Experimental Results and Analysis

In this section, the performance of the proposed DN-MEA method was verified by using the measured data of a corner reflector and UAV. The imaging results were compared with those of the FP-MEA and N-MEA methods. Several SNR conditions were set to verify the robustness of the proposed algorithm.

### 4.1. Experiment Set-Up

In the experiment, we chose a UAV and a corner reflector as the targets. During the radar observation, the corner reflector was led by the UAV to make irregular movements together; radar could track one of the targets in each experiment. The experimental scene is shown in Figure 4. The radar was placed in a proper position to ensure that the UAV was 90 degrees from the radar line of sight (LOS) during flight. The antenna beam was adjusted first before radar tracking to better capture the UAV. During the flight of the UAV, the THz radar antenna received the echo signal and transmitted the echo signal to the computer for processing. The target was then located by analyzing a continuous number of echo signals and tracked by laser equipment. The ISAR image generated in this case inevitably had occlusion effects, such as the fuselage covering a wing.

The imaging THz radar system used in this paper was based on the frequency modulated continuous wave (FMCW) principle. The parameters of the THz radar system in the experiments are listed in Table 2. According to the radar imaging theory, the theoretical value of the range resolution is $\rho =\frac{c}{2B}=5.2\mathrm{mm}$, where *c* and B are the speed of light and bandwidth of the signal, respectively. Figure 5 shows the decibel value image of the normalized one-dimensional range profile of the corner reflector. The corner reflector had a range resolution of 6.1 mm, which was close to the theoretical value, and fully proved the advantages of THz radar in high-resolution imaging, preparing it for subsequent imaging processing. 

### 4.2. Tests and Results with Point Target Imaging 

To verify the reliability of the proposed DN-MEA algorithm, ISAR imaging of the corner reflector was carried out. Figure 6 shows the optical image of the corner reflector. For the purpose of comparison, FP-MEA and the N-MEA were also used for these data. Firstly, the range alignment of the one-dimensional range profile of the target was achieved with the envelope correlation method [34]. Subsequently, the phase errors were compensated by FP-MEA, the N-MEA, and DN-MEA. Finally, the imaging was performed by the range Doppler (RD) algorithm [35].

Figure 7 shows the imaging results of the corner reflectors obtained with FP-MEA, the N-MEA, and DN-MEA, in turn. The imaging results obtained with FP-MEA and the N-MEA show both of the high side lobes that seriously affected the image quality, resulting in fake point targets in the image and affecting the judgment of the real scattering points. The main reason for this was that the entropy function fell into the local optimal solution in the iterative process. While the side lobes could be suppressed by windowing in the fast and slow time domains, windowing would inevitably cause the expansion of the main lobe. In contrast, the proposed DN-MEA algorithm in this study could not only accurately compensate for the phase errors but also achieve side lobe suppression, so that the better peak-focused properties could be significantly presented.

Azimuth resolution and side lobe suppression were analyzed using azimuth profiles. Figure 8 shows the azimuth profiles of FP-MEA, the N-MEA, and DN-MEA. The azimuth profiles were selected to compare −3dB width. Observing the local amplification image in Figure 8b, the widths of the main lobe among these three azimuth profiles were about 1.18, 1.18, and 1 cm in sequence. The azimuth profile view intuitively showed that FP-MEA and the N-MEA prompted high side lobes. In complex target imaging, the main lobe of the weak scattering center might be blocked by the side lobes of the strong scattering center, resulting in the appearance of false points and the loss of some real scattering points. These results were consistent with the theoretical analysis and further demonstrated that the image obtained by DN-MEA had lower side lobes, and improvement was seen on the azimuth resolution.

The peak side lobe ratio (PSLR) is the measurement of the imaging radar system’s ability to identify a weak target from a nearby strong one. The integrated side lobe ratio (ISLR) characterizes the ability to detect weak targets in the neighborhood of bright targets. The PSLR and ISLR of the three methods are listed in Table 3. The two metrics of the proposed algorithm, compared with the FP-MEA and N-MEA methods, were reduced by about 2–4 dB, indicating that the algorithm had a better performance.

### 4.3. Tests and Results with UAV Imaging

In this section, we further tested the proposed algorithm using UAV data measured with THz radar. The optical image of the UAV is shown in Figure 9. Moreover, the UAV was 34 cm in length, 35 cm in width, and 16 cm in height.

Again, the FP-MEA algorithm and the N-MEA were used for comparison. The obtained UAV ISAR image is shown in Figure 10, with the dynamic range adjusted to 30 dB. As shown in Figure 10, many high side lobes, which severely blurred ISAR imaging, appeared in the imaging results obtained by FP-MEA and the N-MEA. There exist several irrelevant fake points with FP-MEA, with the energy of the scatterer diffusing in the adjacent units, resulting in the defocusing of the ISAR image. The results similarly showed that the DN-MEA algorithm proposed in this paper achieved better focus performance than FP-MEA and the N-MEA, especially from the scattering points on the wings, which showed that the DN-MEA algorithm had higher compensation accuracy and was more suitable for the THz band.

Similarly, for the purpose of analyzing the azimuth resolution and side lobe suppression of the proposed algorithms, the azimuth profile was selected to compare the −3 dB width. For the convenience of illustration, the scattering point located at the lower left wing was selected as an example to intuitively analyze the focus performance with DN-MEA. Figure 11 shows the azimuth profiles of FP-MEA, the N-MEA, and DN-MEA. The width of the main lobe was about 0.044, 0.029, and 0.018 m in sequence. In radar imaging, if the side lobes are too high, the closely spaced scattering points produce side lobes–main lobes and side lobes–side lobes overlapping effects, resulting in the misrepresentation of the amplitude peaks on the response of the real points. Especially, the side lobes of strong scattering centers causes a misunderstanding of the targets and misalignment of the main lobe, which destroys the image quality. Figure 11 shows that higher side lobes occur when using the FP-MEA imaging method, while the proposed method had narrow main lobes and low side lobes, which was consistent with the visually observed imaging results.

In ISAR imaging, the influence of SNRs on imaging quality must be considered. An effective imaging algorithm under a low SNR is more meaningful for practical engineering applications. The SNR was defined after range alignment of the range profile, that is, the noise was added to f(m,n) of raw phase history, and the SNR is defined as
(21)$$\mathrm{SNR}=10\log_{10}\frac{E_{S}}{E_{N}}$$
where $E_{S}$ is the power of the signal in the datum domain and $E_{N}$ is the power of the additive complex Gaussian noise. Figure 10 shows the imaging results obtained by adding different Gaussian white noises to the range-compressed echoes. As can be seen from Figure 8, the defocusing of FP-MEA and the N-MEA became more severe at a low SNR. Moreover, when the SNR is as low as −10 dB, a few irrelevant false points appeared in the reconstructed results generated by the N-MEA. The reason lied in that in the N-MEA, a simplified Hessian matrix of image entropy was used, and no search step has ever been applied in the N-MEA to ensure the continuing decline of image entropy, which cannot guarantee the image entropy to converge to optimum solution in a low SNR condition. The autofocusing algorithm based on the global optimal modeling proposed in this paper was more robust against noise, so that the well-focused ISAR images were obtained with a less noisy background at a low SNR. 

Image contrast is similar to image entropy, which is widely used to estimate the focused quality of ISAR images. The image entropy is defined as Equation (8), and the image contrast is expressed as follows:(22)$$C_{I}=\sqrt{\frac{MN}{S}\sum_{k=0}^{M-1}\sum_{n=0}^{N-1}{\left|I(k,n)\right|}^{4}-1}$$

The image entropy and contrast of the radar images shown in Figure 10 and Figure 12 are displayed in Table 3 to quantitatively compare the performance of the three algorithms. The entropy value of the proposed method was the smallest at different SNRs. The image entropy changed very slightly when the SNR was reduced to −10 dB, proving that the algorithm had strong robustness. From Table 4, it can be seen that the proposed DN-MEA method achieved lower image entropy and a higher image contrast than those of either the FP-MEA or N-MEA methods under different SNR conditions, thus verifying its superior performance for different SNRs.

Figure 13 depicts the curves of image entropy and iteration times based on FP-MEA, the N-MEA, and DN-MEA without Gaussian white noise and with an SNR = −10 and −5 dB, respectively. The convergence curve of the FP-MEA algorithm was relatively flat with or without adding noise, and the slope of the curve changed very little, indicating that the convergence was slow. The N-MEA curve appeared to be $E_{I}{}^{\prime }(\varphi_{l+1})>E_{I}{}^{\prime }(\varphi_{l})$, indicating that the search direction determined in the N-MEA did not guarantee a consistent decrease in entropy, which would likely to lead to a situation where the objective function would not converge to a minimal value. The proposed DN-MEA method utilized the golden section method to determine the search step to ensure the convergence of the objective function to the minimum value and converge the image entropy to the optimal level, even under a low SNR condition.

## 5. Discussion

Taking the algorithm’s practicability robustness and accuracy into consideration, a terahertz autofocusing algorithm based on DN-MEA is proposed in this paper. Our method was quantitatively verified by a large number of experimental results based on the measured data of 0.32-THz radar.

Future work can be considered in the following directions: (1) In this experiment, the phase noise of the target echo of the corner reflector as a reference signal was processed. Data-driven phase correction algorithms for terahertz radar systems are an attractive direction, and the proposed method can be considered for the application to correct nonlinear phase errors of terahertz radar systems. (2) In THz-ISAR imaging, the phase error caused by translational motion accounts for a large proportion, while the part of the phase error caused by rotational motion is only a small fraction. Consequently, in this study, only the compensation of the phase error caused by translational motion was considered. However, high-order phase errors caused by rotational motion may also affect the imaging quality, although it inevitably increases the complexity of the algorithm. Therefore, a joint approach of translational and rotational phase error corrections in the terahertz band is a considerable direction. (3) ISAR scaling enables the user to estimate the size of the target from the radar image. In this study, THz-ISAR images were scaled along range cells with a range resolution of c/2B, so the size of the target in the range direction could be estimated. The cross-range scaling needs to estimate the rotation parameters of the target, which is more difficult and is our future research direction.

## 6. Conclusions

In this paper, a minimum entropy autofocusing algorithm based on the damped Newton method was proposed to obtain well-focused and fast-converging ISAR images from a THz radar system with a large datum volume and high sampling rate. This method used the phase error of each pulse as an unknown variable to perform an iterative search. The search step size factor was introduced to ensure the stable decline of image entropy. It can be seen from the experimental results that a reasonable step-size parameter can make the ISAR image converge to a position close enough to the minimum entropy, so as to obtain a well-focused ISAR image. The variation curves of the image entropy functions of FP-MEA, the N-MEA, and DN-MEA with the number of iterations were analyzed for different SNR conditions. The results showed that the proposed method could still generate well-focused ISAR images, even when the SNR was as low as −10 dB, which verified the robustness of the method. Further research is planned to try to incorporate this method into other steps of THz-ISAR imaging.

## Figures and Tables

**Figure 1 sensors-22-06883-f001:**
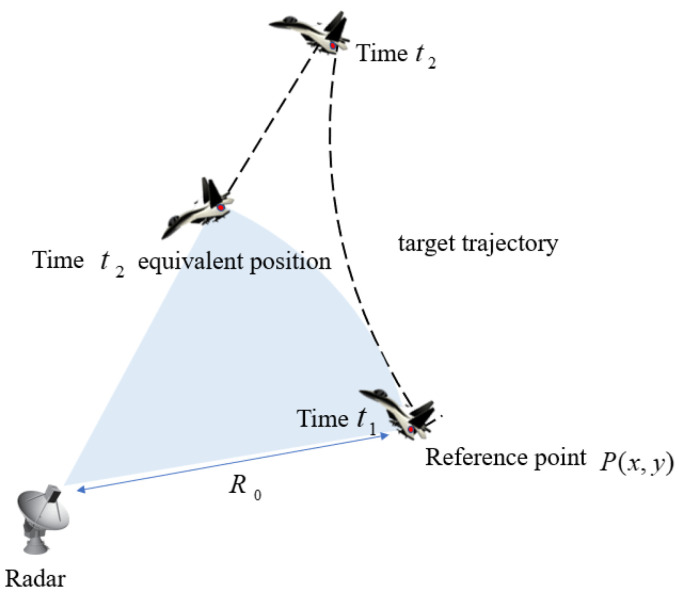
The equivalent diagram of ISAR translational motion.

**Figure 2 sensors-22-06883-f002:**
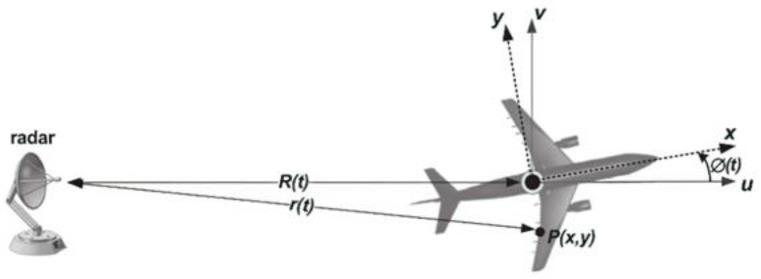
Geometry for a moving target with respect to radar.

**Figure 3 sensors-22-06883-f003:**
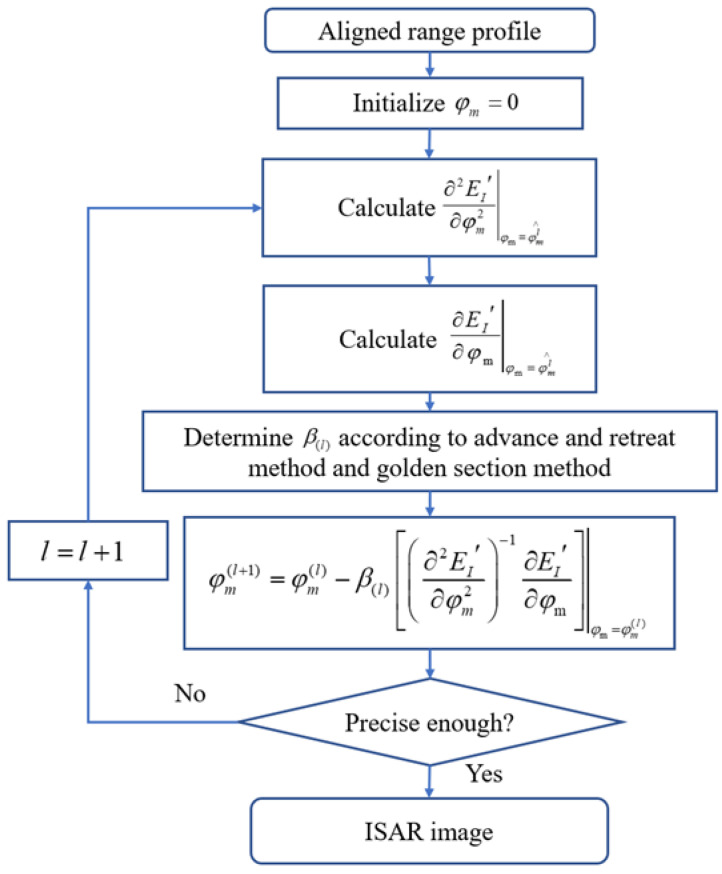
Flow chart of DN-MEA.

**Figure 4 sensors-22-06883-f004:**
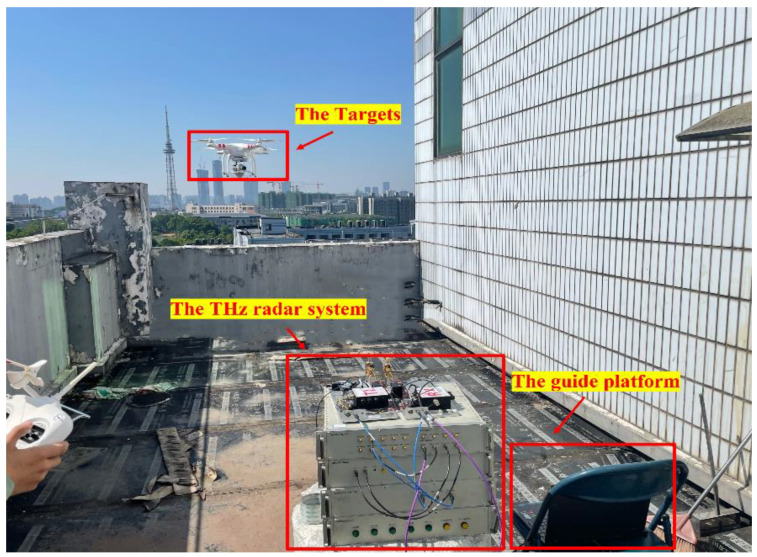
The experiment scene.

**Figure 5 sensors-22-06883-f005:**
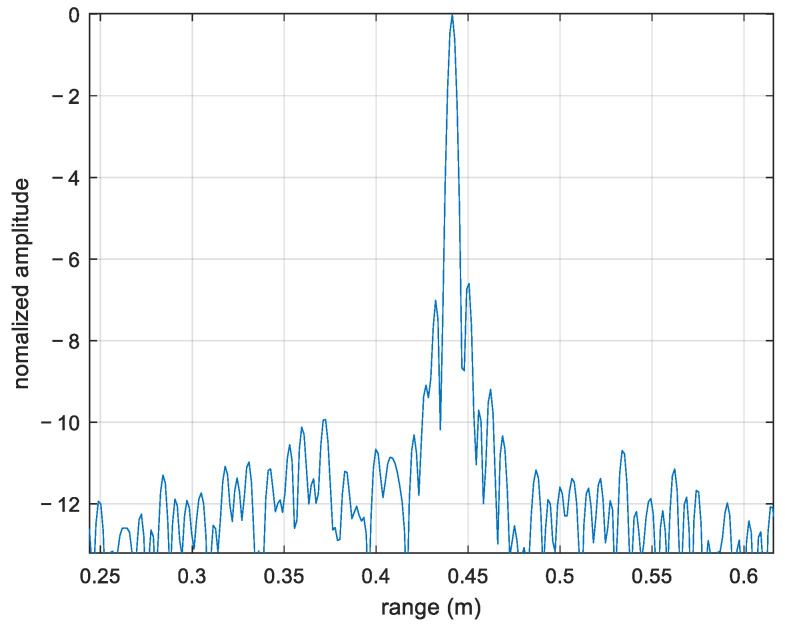
Normalized amplitude decibel value.

**Figure 6 sensors-22-06883-f006:**
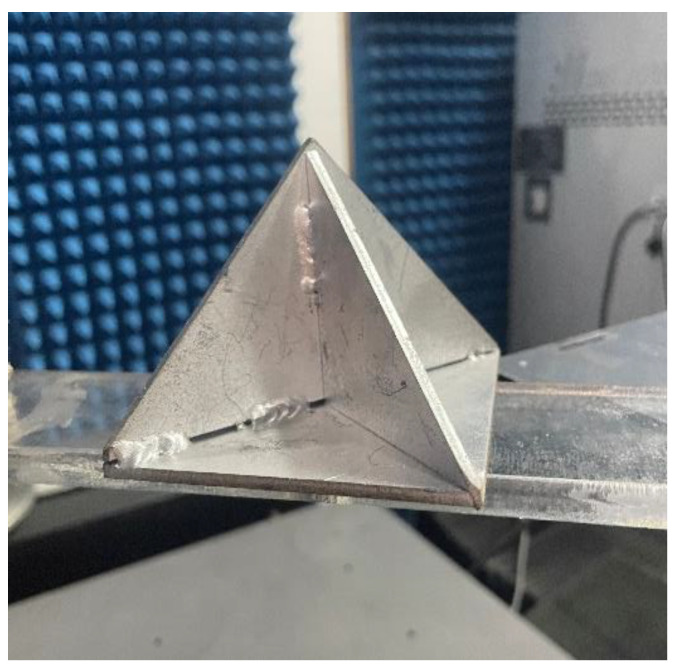
The optical image of the corner reflector.

**Figure 7 sensors-22-06883-f007:**
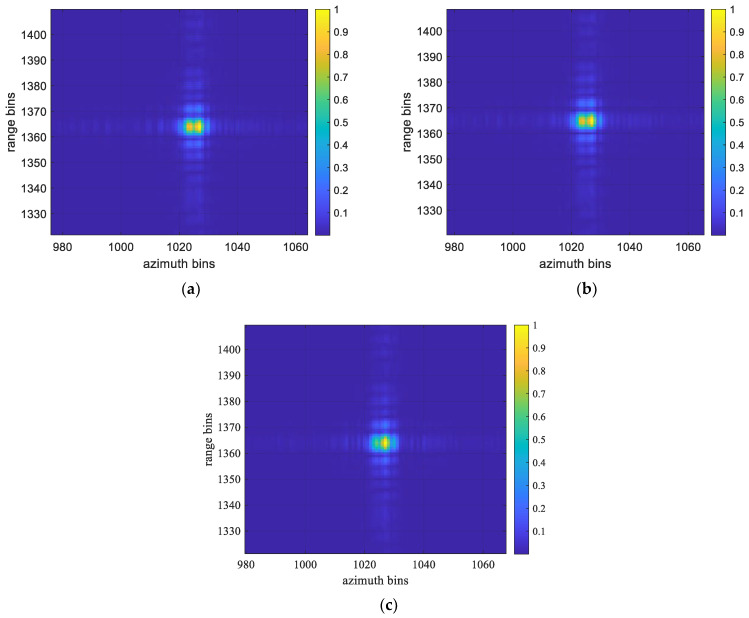
The imaging results of corner reflectors with different algorithms: (**a**) FP-MEA; (**b**) N-MEA; (**c**) DN-MEA.

**Figure 8 sensors-22-06883-f008:**
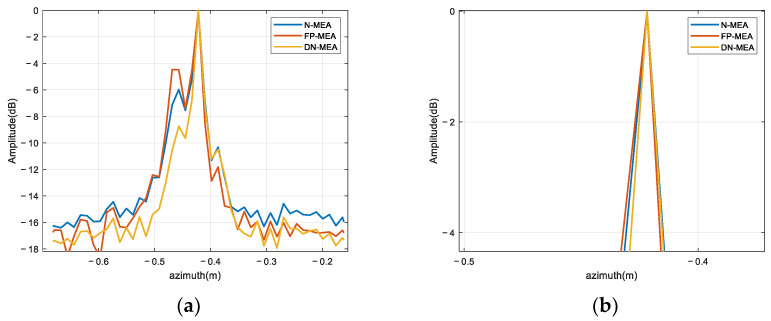
Azimuth profiles of the corner reflector imaging results: (**a**) original results; (**b**) local amplification.

**Figure 9 sensors-22-06883-f009:**
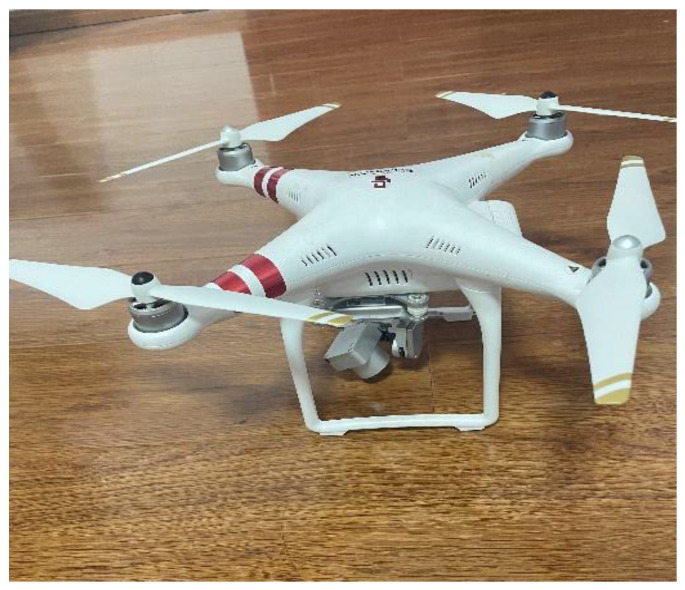
The optical image of the UAV.

**Figure 10 sensors-22-06883-f010:**
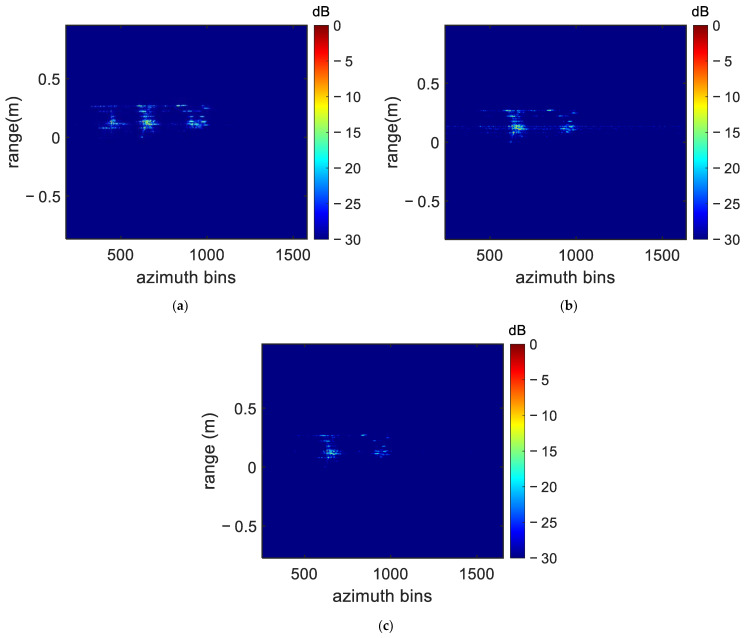
The imaging results of different algorithms: (**a**) FP-MEA; (**b**) N-MEA; (**c**) DN-MEA.

**Figure 11 sensors-22-06883-f011:**
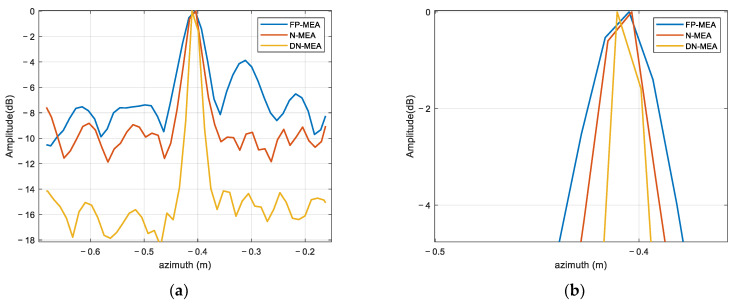
Azimuth profiles of the target imaging result: (**a**) original results; (**b**) local amplification.

**Figure 12 sensors-22-06883-f012:**
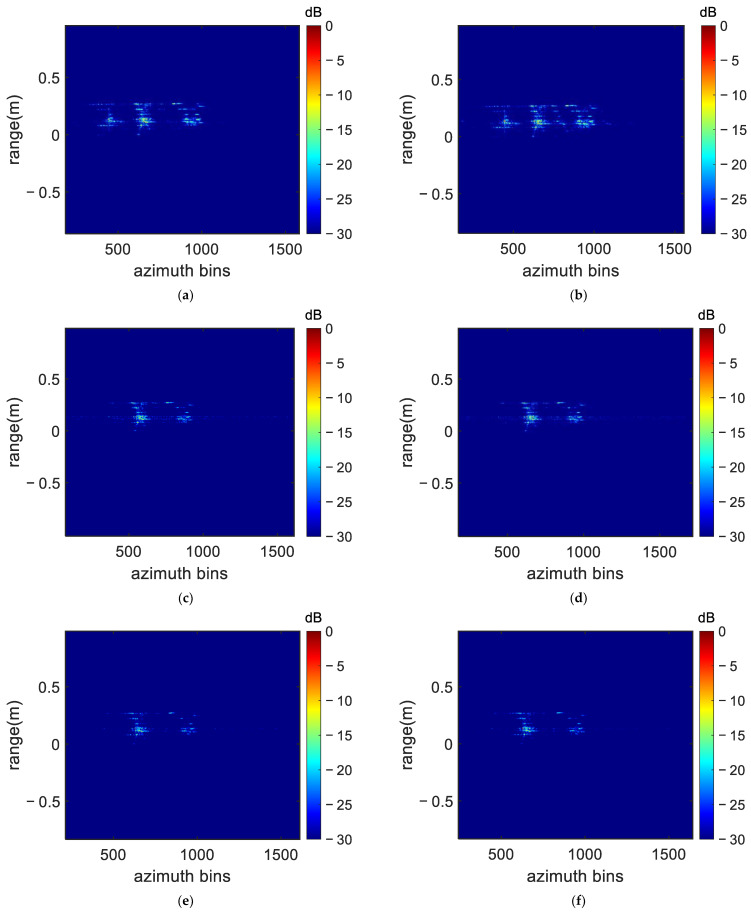
The imaging results of different methods in different SNR conditions: (**a**) FP-MEA (−5 dB); (**b**) FP-MEA (−10 dB); (**c**) N-MEA (−5 dB); (**d**) N-MEA (−10 dB); (**e**) DN-MEA (−5 dB); (**f**) DN-MEA (−10 dB).

**Figure 13 sensors-22-06883-f013:**
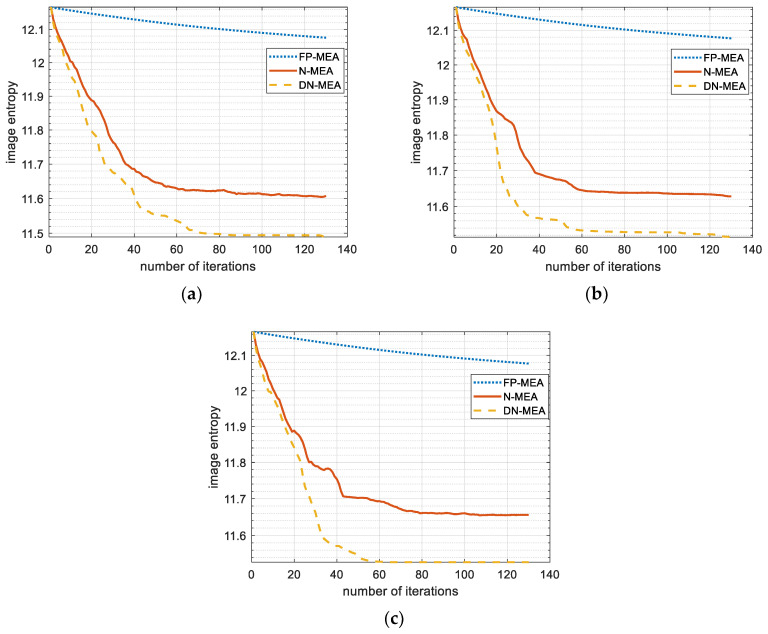
Curves of image entropy versus the number of iterations of different methods and SNR conditions: (**a**) without the addition of noise; (**b**) −5 dB; (**c**) −10 dB.

**Table 1 sensors-22-06883-t001:** Comparison of several popular motion compensation methods.

Method	Robustness	Precision	Computational Efficiency	Adaptability
Parameterization based on heuristic search [31]	strong	low	high	weak
Joint envelope alignment and phase average compensation [26,32]	weak	high	average	average
Regularization and sparse class methods [33]	average	high	high	average
Image-quality-measurement-based methods [24,25]	strong	average	low	strong

**Table 2 sensors-22-06883-t002:** Parameters of the radar system.

Parameter	Symbol	Value
Carrier frequency	*f_c_*	0.32 THz
Bandwidth	*B*	28.8 GHz
Pulse width	*T_p_*	0.3 ms
IF sampling frequency	*f_s_*	1.56 MHz
Pulse repetition frequency	*PRF*	1000
Pulse number	*N*	1000
Pulse sampling points	*M*	2048
Range resolution	*ρ*	5.2 mm

**Table 3 sensors-22-06883-t003:** Quality assessment metrics.

Metrics	FP-MEA	N-MEA	DN-MEA
ISLR	−3.4549	−4.2243	−5.0657
PSLR	−4.7789	−5.9788	−8.7456

**Table 4 sensors-22-06883-t004:** Image entropy and image contrast of imaging results in Figure 6 and Figure 8.

Algorithms	Indicators	Original Data	−5 dB	−10 dB
FP-MEA	Image entropy	11.7602	11.7887	11.8033
	Image contrast	17.2855	16.9824	15.5231
N-MEA	Image entropy	11.6461	11.6823	11.7056
	Image contrast	22.8327	21.8398	19.6554
DN-MEA	Image entropy	11.5264	11.5319	11.5384
	Image contrast	29.5762	28.0903	26.0834

## Data Availability

Not applicable.

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
