# Peer review of "Autofocusing of Maneuvering Targets in Terahertz Inverse Synthetic Aperture Radar Imaging Based on Damped Newton Method"

_sensors, 2022, doi:10.3390/s22186883_

Round 1

Reviewer 1 Report (New Reviewer)

In this work, authors have proposed a MEA algorithm based on the Damped 16 Newton method which improves computational efficiency by approximating the first- 17 order and second-order partial derivatives of the image entropy function with respect to the phase 18 errors as claimed by the authors. The manuscript is written well. I have few suggestions for the authors: 

1) In the introduction section, the significance of ISAR is not established well. The author needs to mention some of the key applications of ISARs.

2) The author needs to mention the major contributions in the current work in the introduction section. 

3) If the authors provide the tabular comparison of some previous works on ISAR with the current work, this will further increase the significance of current work. 

4) The author needs to mention some more recent works specifically from 2019-2022. 

Author Response

Reviewer 2 Report (New Reviewer)

[1] Eqn.(4): Missing time derivative.

[2] Fig.12: Please elaborate how the noise is added to the measured signal to render specific SNR level.

[3] Line 345: Please elaborate why the translational motion error is the major error source that is compensated with the proposed method. This will help highlight the significance of this work.

Author Response

Reviewer 3 Report (New Reviewer)

This paper presents an autofocusing method for imaging maneuvering targets using the terahertz ISAR technique. 

The paper is well-written. The novelty and effectiveness of the proposed method are clearly presented. 

Thus, I think this paper can be accepted for publication. 

Author Response

Thank you so much for acknowledging our work.

This manuscript is a resubmission of an earlier submission. The following is a list of the peer review reports and author responses from that submission.

Round 1

Reviewer 1 Report

The article was well written, but all the figures need to be reviewed and the discussion and conclusions needs to be reviewed and more elaborated.

Se below some reviews that need to be considered by the authors:

Line 16: The meaning of THz needs to be inserted in the abstract.

Line 210: In the equation of range resolution the meaning of “c” and “B” needs to be explained.

Line 224: In the Figure 5 (b), what is the unit for “Azimuth” and the unit for the color scale from 0 to -30?

Line 237: The Figure 6 needs to be reviewed. 

Line 254: The Figure 8 needs to be reviewed and for a good comparison all graphics could be presented in the same page.

Line 281: The Figure 10 needs to be reviewed and for a good comparison all graphics could be presented in the same page.

Line 307: The “Discussion” is too short and very similar to the “Conclusions”. I think that “Discussion” need to be reviewed to reflect the results of the work in deep.

Reviewer 2 Report

Please find the attached comment

stay safe,

Round 2

Reviewer 1 Report

No comments for the authors.

Reviewer 2 Report

Despite all efforts, the author's descriptions have exacerbated the understanding of their proposed method rather than solved it! The Autofocusing concept is claimed to be distinct from SAR, despite that, their  description, classification and approach are identical to SAR! The proposed method has been termed "Autofocusing" in ISAR imaging, while it is mostly concerned with 2D image pixel quality enhancement rather than focusing (according to the author's comments in 4-8, 4-9, and 4-12)! In short, even if the goal is to improve image quality, it must contribute to the body of knowledge, which is not the case here!  Again, the same old problems exist (mostly regarding the formulation, geometry and quality assessment). I think authors must start over.

As a result, I would reject it and would not advise resubmitting it.

Stay safe,